# Ferroelectric Thin Films and Composites Based on Polyvinylidene Fluoride and Graphene Layers: Molecular Dynamics Study

Vladimir Bystrov [1,*], Ekaterina Paramonova [1], Xiangjian Meng [2], Hong Shen [2], Jianlu Wang [2], Tie Lin [2] and Vladimir Fridkin [3]

[1] Institute of Mathematical Problems of Biology—Branch of Keldysh Institute of Applied Mathematics, Russian Academy of Sciences, 142290 Pushchino, Russia; ekatp@yandex.ru

[2] National Laboratory Infrared Physics, Shanghai Institute of Technical Physics, Chinese Academy of Sciences, Shanghai 200083, China; xjmeng@mail.sitp.ac.cn (X.M.); hongshen@mail.sitp.ac.cn (H.S.); jlwang@mail.sitp.ac.cn (J.W.); lin_tie@mail.sitp.ac.cn (T.L.)

[3] Federal Center of Photonics and Crystallography, Shubnikov Institute of Crystallography, Russian Academy of Sciences, 119333 Moscow, Russia; fridkinv@gmail.com or fridkin@ns.crys.ras.ru

* Correspondence: vsbys@mail.ru or bystrov@impb.ru

**Abstract:** This work is devoted to the study of nanosized polymer polyvinylidene fluoride (PVDF) thin ferroelectric films (two-dimensional ferroelectrics) and their composites with graphene layers, using molecular dynamics methods to (1) study and calculate the polarization switching time depending on the electric field and film thickness, (2) study and calculate the polarization switching time depending on changes of the PVDF in PVDF-TrFE film, and (3) study the polarization switching time in PVDF under the influence of graphene layers. All calculations at each MD run step were carried out using the semi-empirical quantum method PM3. A comparison and analysis of the results of these calculations and the kinetics of polarization switching within the framework of the Landau–Ginzburg–Devonshire theory for homogeneous switching in ferroelectric polymer films is carried out. The study of the composite heterostructures of the "graphene-PVDF" type, and calculations of their polarization switching times, are presented. It is shown that replacing PVDF with PVDF-TrFE significantly changes the polarization switching times in these thin polymer films, and that introducing various graphene layers into the PVDF layered structure leads to both an increase and a decrease in the polarization switching time. It is shown that everything here depends on the position and displacement of the coercive field depending on the damping parameters of the system. These phenomena are very important for various ferroelectric coatings.

**Keywords:** molecular dynamics; quantum-chemical semi-empirical methods; Landau–Ginzburg–Devonshire theory; polarization; nanoscale ferroelectrics; polymers thin films; PVDF; PVDF-TrFE; homogeneous switching; switching time; coercive field; graphene; composites; heterostructures

## 1. Introduction

Polymer ferroelectric polyvinylidene fluoride (PVDF) and its copolymer with trifluoroethylene (PVDF-TrFE) are well-known materials with excellent piezoelectric properties, mechanical and thermal stability, and constantly growing areas of application [1–14], including energy harvesting devices, in sensor and actuator material mechanisms, in biomedical and other devices, and in the field of flexible organic electronics/nanoelectronics as layers and coatings for various heterostructures. Due to their flexibility, thin films of these ferroelectric polymers make it possible to create piezoelectric composite materials for use in flexible piezoelectric nanogenerators [7]. These include piezoelectric sensors integrated into clothing, which represents great potential for future wearable electronics, and an approach for fabricating flexible piezoelectric sensors based on PVDF/graphene composite coatings

on commercially available fabrics has also been reported [8]. Moreover, various modifications of graphene (G), graphene oxide (GO), reduced graphene oxide (rGO) [9,10], as well as the use of these materials to create nanocomposites with various polymer matrices, are being studied [11,12]. Based on polystyrene graphene foam (PSGF) on a PVDF substrate, their composition has been proposed as an efficient current collector and for lithium metal anodes (to improve their lithium battery performance) [13].

Another direction of application of thin ferroelectric layers based on PVDF and P(VDF-TrFE) films is the production of metal-ferroelectric-semiconductor field-effect transistors (MFSFETs) [14]. The characteristics of such MFSFETs made from thin PVDF and P(VDF-TrFE) films have been shown to have very good ferroelectric hysteresis curves with a counterclockwise loop, the same as other ferroelectric materials.

In recent years, due to the rapid development of nanoelectronics, new organic devices based on PVDF and PVDF-TrFE thin films have attracted intense research interest. Here, along with experimental studies, theoretical approaches are widely used, including the use of computer modeling and calculations using various numerical methods. For the theoretical study of such thin ferroelectric nanofilms and the modeling of polarization switching processes in them, modern methods of molecular dynamics (MD) [15–17], in combination with the use of computer molecular modeling and quantum chemical calculations, are very well suited.

This paper discusses computer modeling methods and especially the application of the MD simulation method to study the properties of nanomaterials that are promising for many applications, such as nanoscale ferroelectrics [18–22]. These are, first of all, nanomaterials based on a polymer structure (PVDF) and its copolymer P(VDF-TrFE), especially those made using the Langmure–Blodgett (LB) technique [23–25], as well as its composites with graphene (G) [26–28]. It is important that these theoretical studies are carried out in close combination with experimental studies.

The idea is that these ferroelectrics demonstrate a striking finite-size effect [19]. The switching time and coercive field increase with decreasing film thickness, while the polarization and phase transition temperature decrease with decreasing thickness [20,21]. This is important for applications that require operation at low voltage, for example in devices such as non-volatile memory, where ferroelectric films must be quite thin.

The discovery of Langmuir–Blodgett (LB) ferroelectric polymer films [23] led to new studies of ferroelectric properties at the nanoscale (one to two monolayers thick (ML, 0.5–1.0 nm)) [19–25]. This has opened the way for studying the finite-size effect at the nanoscale, which is also important in many technical applications. The results obtained on P(VDF-TrFE) LB films open new directions in fundamental physics, such as the existence of two-dimensional ferroelectrics [20,25] and the transition from domain to homogeneous polarization switching in the context of the Landau–Ginzburg–Devonshire (LGD) mean field theory for thicknesses less than 15–20 nm [18–25].

It is extremely important for the development of further practical applications of such ferroelectric thin layers and coatings based on PVDF and PVDF-TrFE, including their composites with graphene (PVDF-G) and graphene-like layers [28], to clarify the theoretical basis of the physical processes in these thin ferroelectric layers and the processes of polarization switching, which are important for nanoelectronics and related fields.

To study the properties of such ferroelectric nanomaterials and nanocomposites, in addition to the experimental methods, various methods of computer simulation and numerical quantum-chemical studies are also used [29–33].

In this work, we applied MD simulation methods, in which the calculations of the interaction of the atomic–molecular system of the nanomaterials were carried out on the basis of semi-empirical quantum-mechanical (QM) calculations at each step of the MD run, using methods such as AM1, PM3, etc., [18,33–38], as presented in the HyperChem software (Versions 8.0/01,) [38]. We have already carried out these calculations for such polymer ferroelectrics PVDF and P(VDF–TrFE) [39–41]. These QM methods during MD

runs were used to calculate the polarization switching time depending on the magnitude of the electric field [18,21,22,42].

At present, the creation of new composite materials based on PVDF, PVDF-TrFE, and graphene with improved characteristics is of great interest [18–21,43,44]. MD techniques here can enhance the understanding of the interactions between graphene layers and polymer ferroelectric layers.

In addition to PVDF and PVDF-G, this work also examines in more detail PVDF-TrFE and copolymer composite structures such as PVDF-TrFE-G and G-PVDF-TrFE-G and calculates the polarization switching times in them via MD methods using semi-empirical quantum calculations at each step of the MD run [41,42].

This article is devoted to new calculations of polarization switching times using MD methods with quantum calculations at each step; discussion and analysis of these calculation results for the composite heterostructures PVDF and PVDF-TrFE with different G-layers, both in experiment and in simulation; and further prospects for studying their properties using these methods.

## 2. Basic Models

First, we consider the main experimental data and models of PVDF and P(VDF-TrFE) ferroelectrics and thin ferroelectric films, as well as the theoretical foundations and estimates of the polarization switching times in them. Then, we also consider calculations of the polarization switching time using the methods of molecular MD runs.

### 2.1. Nanoscale Two-Dimensional and Polymer Ferroelectric Thin Films

Ferroelectric single-crystal thin films based on PVDF/P(VDF-TrFE) were obtained for the first time in the early and mid-1990s using the Langmuir–Blodgett (LB) technique [19,23–25], based on the transfer of chains and monolayers (MLs) of the polymer (formed on the surface of water in a special bath) to the substrate carrying the electrodes.

Figure 1a–d shows the models of PVDF polymer chains in polar (ferroelectric) and non-polar (paraelectric) phases and the corresponding cells of their crystal structure (in sections across the polymer axis). These model images were built using HyperChem [38] and are widely used in our works [21,22,28,39–42]. Figure 1e,f shows a transferring scheme of LB PVDF layers and chain patterns of one ML observed in a tunneling microscope (Figure 1g).

LB ferroelectric films obtained by this method in [23–25] turned out to be of record thinness. These nanosized two-dimensional ferroelectrics (with a thickness of 0.5–1.0 nm) consisting of a single monolayer were obtained in [24] with polarization at the order of P~0.1 C/m$^2$ [23–25]. The thickness of these LB polymer films (two-dimensional ferroelectrics) was controlled by ellipsometry and atomic force spectroscopy [23–25,45]. The thickness of one monolayer (1 ML) was L = 0.5 nm = 5 Å, i.e., much less than the theoretical estimate of the size of the ferroelectric critical domain nucleus $x_0$~10...20 nm (L << $x_0$) known from the literature [19–21,46]. The discovery of nanosized two-dimensional polymer ferroelectrics [23–25] led to a new stage of ferroelectric properties, including the study of switching in ultrathin films, using the MD method [18]. The LGD theory is well suited for describing ferroelectric phenomena in such nanosized polymer ferroelectrics [19–25]. This LGD approach was proposed firstly in work [47–50] and developed in [51–54], including further experimental investigations. We discuss all these issues more detailed in below sections.

### 2.2. Main Models of Polymer Ferroelectric Thin Films

Polymer ferroelectric films based on PVDF and P(VDF-TrE) have also been reported [19,22]. These were the bulk ferroelectric spun films formed by solvent techniques; they were usually made by the spin-coating method [55–57]. They possess main ferroelectric properties similar to the thin LB films [19–25], but unlike the thin two-dimensional LB films they are thicker and volumetric (three-dimensional bulk).

Besides this, these spun ferroelectric polymer films are not (and cannot be) as well structurally ordered as LB films, and they have many structural inhomogeneities, deposits of amorphous phases and polycrystals, and completely different (domain) polarization-switching mechanisms [46].

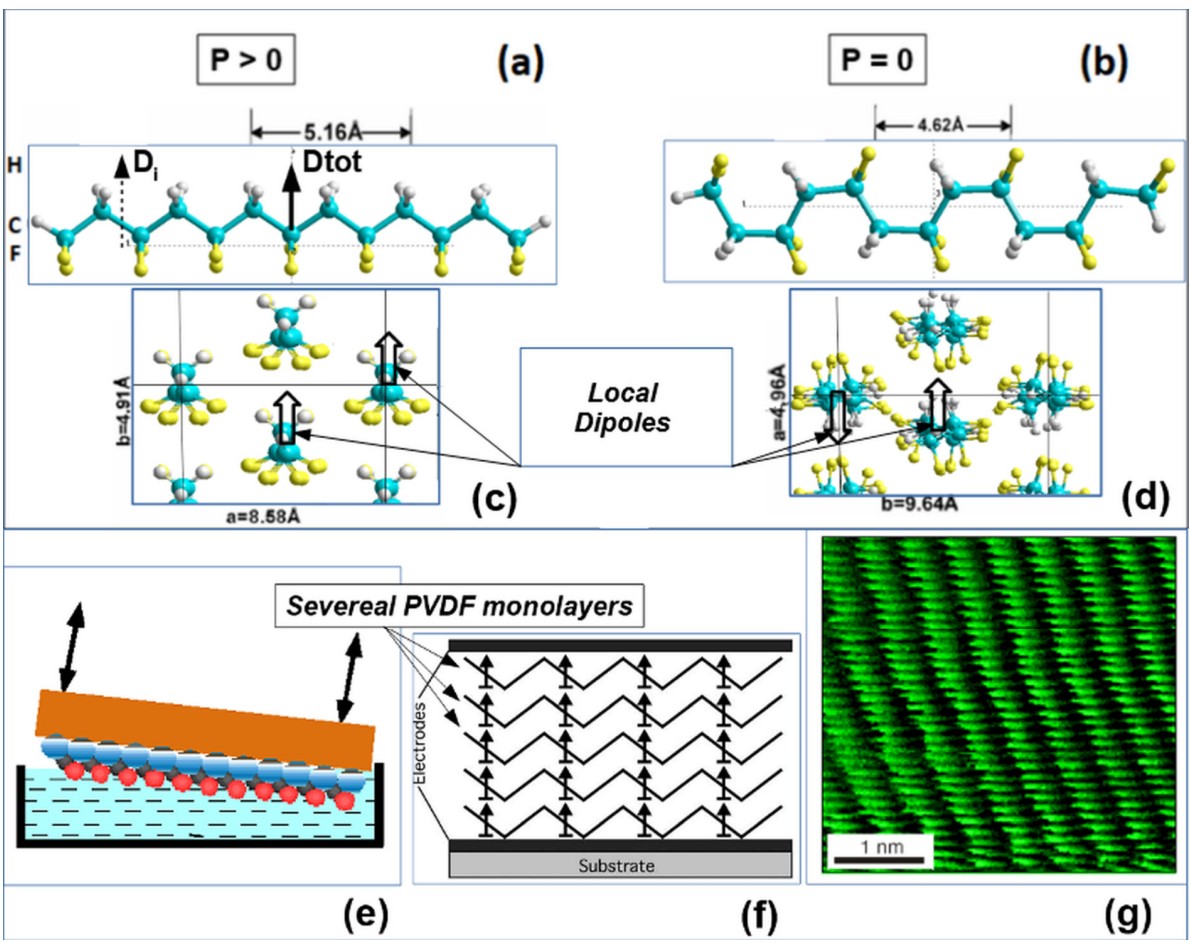

**Figure 1.** Ferroelectric polymer polyvinylidene fluoride (PVDF): (**a**,**c**) PVDF in polar trans conformation and total polarization $p > 0$. (**b**,**d**) PVDF in a nonpolar Gauche conformation with total polarization $p = 0$ (the arrows show the direction of the local dipoles vectors). (**e**) The formation of the PVDF Langmuir–Blodgett (LB) film on the surface of water. (**f**) Transferring several monolayers (MLs) of LB PVDF film onto a substrate with an electrode. (**g**) Image of the 1 M LB film of polyvinylidene fluoride-trifluoroethylene P(VDF-TrFE), obtained via scanning tunneling microscopy. (Reprinted with permission from [21]).

Nevertheless, in their main polar β-phase, they are similar to LB films in their chemical composition and organization. In its β-phase, PVDF has a spontaneous polarization $P$ of about 0.08–0.1 C·m$^{-2}$ [19,22]. Similar properties can be obtained in thin polymer films fabricated by the LB technique—in β-phase structures, this LB film with the dipoles of all PVDF chains is high-ordered aligned in a structure (Figure 1a) with a maximum polarization of $P \sim 0.13$ C·m$^{-2}$ [19,22–25].

It is important to note that ferroelectric polymers based on PVDF and P(VDF-TRFE) turn out to be a convenient object for molecular modeling: they have a sufficiently clear structure of polymer chains, consisting of periodically repeated $C_2H_2F_2$ units with a dipole moment (Figure 1a). Testing of the PVDF models with different numbers of monomer units [39,40] showed that a PVDF model with six $C_2H_2F_2$ units (PVDF6 model) is comparable to a PVDF chain with 10 monomers. Thus, this model is sufficient for all necessary calculations, and this PVDF-6 model has been used primarily in this work.

## 3. Main Methods

### 3.1. Main Theoretical Approach

The theoretical description of ferroelectrics was developed by Ginzburg [47,48] and then by Devonshire, based on Landau's theory of phase transitions [49]. This phenomenological LGD thermodynamic theory was the main basis for the development of the physics of ferroelectrics [19,22]. The LGD theory explained all the basic properties of ferroelectrics, including polarization switching in an external electric field and the hysteresis loop. However, it turned out that it does not describe all the switching phenomena of a ferroelectric, since it predicts the magnitude of the coercive field to be 2–3 orders of magnitude larger than the experimental one. The large coercive fields following from the LGD theory are internal (intrinsic or proper), while the measured experimental values are external (extrinsic or improper). It is significant here that the LGD theory considered the ferroelectric crystal as a homogeneous infinite medium.

As has been established, in conventional ferroelectrics it turns out to be important to split the crystal into domains with a critical size $x_0 \sim 10$–$20$ nm, which leads to such a decrease in the measured external coercive fields [46].

The study of nanosized two-dimensional ferroelectrics based on PVDF and then on the basis of other materials showed that in these nanomaterials, ferroelectricity exists even at sizes much smaller than $x_0 \sim 10 \ldots 20$ nm (at least in the thickness of one monolayer of such materials), as in a continuous homogeneous media. In this case, it is precisely the homogeneous domainless switching of polarization that occurs here immediately in the entire volume of such a nanosized ferroelectric in full accordance with the LGD theory (within size $L \leq x_0$, the system is homogeneous). Then, the LGD theory turns out to be applicable and completely valid. This has been convincingly demonstrated in recent works [20,21] on the analysis of the switching phenomena in such nanosized two-dimensional ferroelectrics. Therefore, here we justifiably rely on this LGD theory when considering ferroelectric phenomena in thin polymer materials based on PVDF/P(VDF-TrFE).

In ferroelectrics, polarization $P$ is the main order parameter (or ordering) of the system. Therefore, according to the Landau theory [49] and its development in the LGD theory [47,48], the behavior of a system (ferroelectric) in the vicinity of a phase transition (PT) between the paraelectric and ferroelectric phases can be described by expanding the thermodynamic potential $F$ (or Gibbs free energy density) in the even degrees of the spontaneous polarization $P$ (in the simple uni-axial case) [19–22]. At the same time, the polarization switching kinetics of two-dimensional polymer ferroelectrics has been described by the Landau–Khalatnikov equation [50], and its solution for first-order phase transitions in two-dimensional ferroelectrics was considered in [51,52]. An investigation of the solution of this equation showed that in the vicinity of the coercive field $E_C$ (when $E \rightarrow E_C$ and $E > E_C$), the switching time $\tau$ increases abruptly and its dependence from $E$ can be expressed [51,52] as

$$\frac{1}{\tau_s^2} = \frac{1}{\tau_0^2}\left(\frac{E - E_C}{E_C}\right), \tag{1}$$

$$\tau_0 = 6.3\xi\gamma\beta^{-2}. \tag{2}$$

where $\xi$ is the damping coefficient.

In this case, $E_C$ is the proper (intrinsic) coercive field of the ferroelectrics. This relation (1) shows directly the linear behaviour of $\tau^{-2}$ along $E$ in the vicinity of $E_C$. This relation is more suitable for comparison with experimentally measured data and is used in [19,53]. Further, this relationship turns out to be convenient for analyzing the results of theoretical calculations when modeling the polarization switching processes in polymer ferroelectrics and allows us to perform this investigation via MD methods [18,20,21,40,42]. These results were also well confirmed by experimental studies [54], which turned out to be consistent with the calculation data [20,21].

### 3.2. Main Computational Details pf MD Run Method for Thin Polymer Ferroelectric Films

For quantum-mechanical calculations, a semi-empirical method of molecular orbitals with a self-consistent field [31–38] was applied in the PM3 parameterization in the Hartree–Fock approximation, similar to AM1 [38]. The approach was developed in detail and tested by Stewart [34–37], and the effectiveness of its use on systems of similar organic polymers was also shown. In [18,20,21,42], the HyperChem software package [38] was used, containing all the above-mentioned necessary methods. In addition, this package contains the necessary means of optimizing the system: to search for a minimum of the system energy, the method of conjugate gradients (Polak–Ribiere) is used [38]. The HyperChem package implements special software for molecular dynamics and the ability to perform MD calculations (runs) using different methods at each step of the MD run, and this software allows us to simulate the application of an external electric field. Performing such MD runs requires specifying a set of parameters, which are indicated in the special molecular dynamics options of the HyperChem program [38]. The main parameters for the considered PVDF model and other related models are as follows: (1) MD calculations at a constant temperature in vacuum, (2) parameter of the "bath relaxation" time = 0.05 ps, (3) MD "run time" = 5–20 ps (for various applied electric field values), and (4) step size" Dt = 0.0005–0.001 ps. These calculations were performed when the electric field $E$ changed in the order of ~(0.001–0.010) a.u.~(0.514–5.14) GV/m.

### 3.3. Homogeneous Polarization Switching in Polymer Ferroelectrics by MD Run Method

The MD run using PM3 method calculations at each MD step under different electric fields was firstly proposed and performed in our work [18] for one PVDF chain (Figure 2).

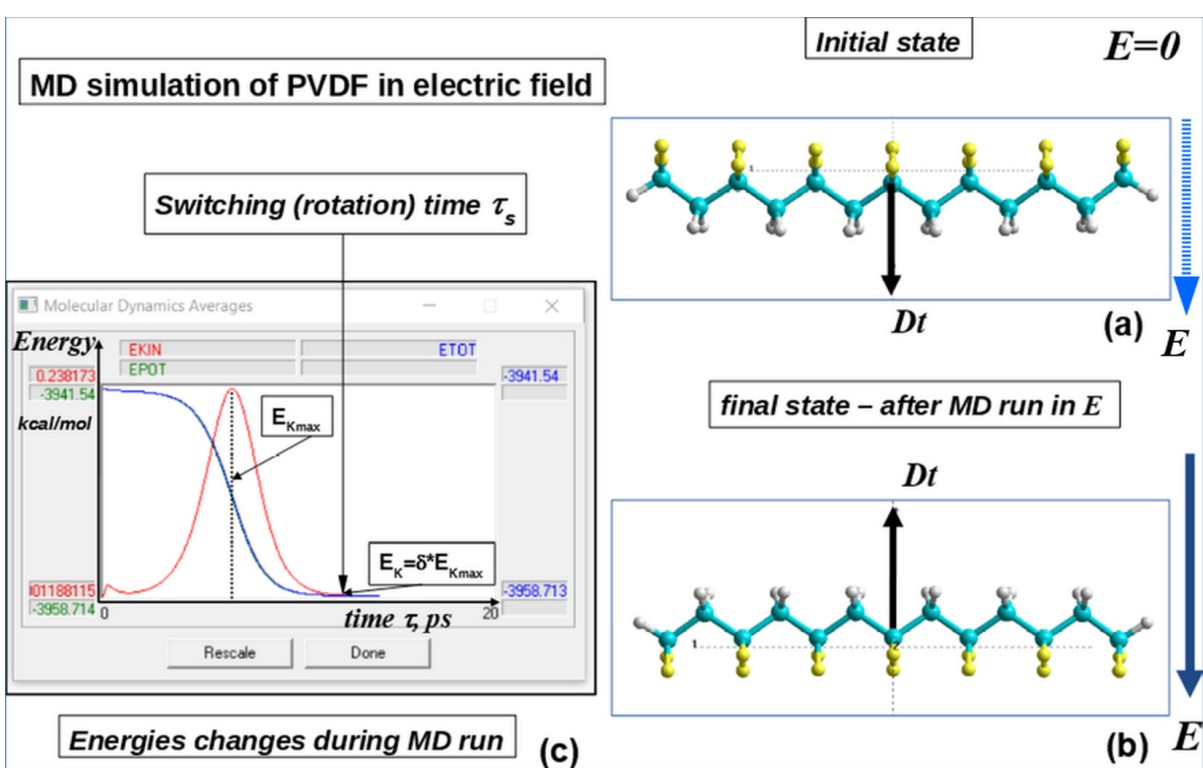

**Figure 2.** Schematic of the MD run process for one PVDF6 chain model using PM3 RHF method calculations in HyperChem software (version 8.0/01?) at each MD run step: (**a**) initial state, (**b**) final state after MD run with rotation (switching) in the opposite direction to dipole moment *Dt* orientation in the electric field *E*. Inset (**c**) show the energy changes during MDS run with respect to time (in ps). (Adapted with permission from [18]; Elsevier, 2014).

Figure 2 shows the case of the PVDF-6 model, which is rotated in the applied electric field E. Here, it is shown how the switching of the direction of the general moment of the dipole moment *Dt* (and the corresponding polarization vector *P*) occurs from the initial direction to the opposite one under the influence of an external electric field.

In this work, the main calculations were carried out using the PM3 quantum method in a restricted Hartree–Fock approximation (RHF) [38] at each step of the MDS run. All calculation results are collected in special numerical files and displayed in the form of trajectories on a special window for MD run data in the HyperChem workspace (Figure 2c). The final time (switching time: $\tau_S$) is estimated from these MD energy trajectories (see Figure 2c) using the following criteria [10]: $\delta = E_K/E_{Kmax} < 10^{-3}$, where $E_K$ is the kinetic energy at the end point, and $E_{Kmax}$ is the kinetic energy at the maximum point EKIN of chain rotation (as shown by the red line in Figure 2c). In fact, this corresponds to the achievement of the rest point of the rotating PVDF chains, when $E_K \sim 0$, and its position with an opposite orientation of the total dipole vector *Dt* in relation to the initial one and the corresponding polarization vector *P*. These data for $\tau_S$—polarization switching times obtained in MD runs at different values of the applied field E—are then used to plot the $\tau(E)$ and $\tau^{-2}(E)$ dependencies according to Equation (1).

## 4. Main Results

### 4.1. Polarization Switching of PVDF Chain

The results obtained in the time calculations of the switching time $\tau$ with different values of the electric field $E$ (performed for the first time in [18]) shows the validity of the polarization switching homogeneous kinetics in accordance with the LGD theory and Landau–Khalatnikov equation [50–52]. This produces linear behavior of the reverse square of the switching time $\tau^{-2}$ from electric field $E$ in Equation (1) for $E$, when $E \to E_C$ ($E > E_C$), where $E_C$ is the boundary value of the coercive field. Such behavior of this dependence of the $\tau^{-2}$ on E (by MD run), in a logarithmic scale, turns out to be in line with the experimentally observed $\tau^{-2}(E)$ dependencies on the thin ferroelectric LB films in [18,20,21,53,54].

### 4.2. Polarization Switching in a Heterostructure Consisting of PVDF and Graphene Layers

4.2.1. Main Details

Further, we examined the influence of graphene layers on important properties of PVDF polymer films, such as polarization switching. To assess the influence of graphene on the polarization switching times in a heterostructure, consisting of a polymer ferroelectric PVDF and graphene, the above-considered model of PVDF6 and MD run methods are used here [18,20,21], as discussed above in Section 3.1.

To model graphene layers, molecular models from [28,41,42], consisting of the 54 carbon atoms C surrounded at the edge by hydrogen atoms H—model Gr54H—are used (Figure 3a,b). Two types of the heterostructure model are considered here as models of the main composite heterostructures of a ferroelectric polymer with graphene:

(1) One-sided model of a PVDF chain and a graphene layer PVDF6 + Gr54H_H-C, where the PVDF chain (or layer) is oriented towards the graphene layer by hydrogen atoms H (Figure 3c).

(2) A double-sided model (or sandwich model), consisting of a PVDF chain enclosed between two layers of graphene Gr54H + PVDF6 + Gr54H (Figure 3d).

4.2.2. Results

Similarly to the procedures of MD runs with models of pure PVDF chains, MD runs of composite models of PVDF with graphene, presented in the initial states in Figure 3c,d, were carried out. As a result, all initial models of the system undergo various structural changes under the action of applied electric fields E of various magnitudes—here, the PVDF dipole structure rotates at different speeds, depending on the magnitude of the applied electric field.

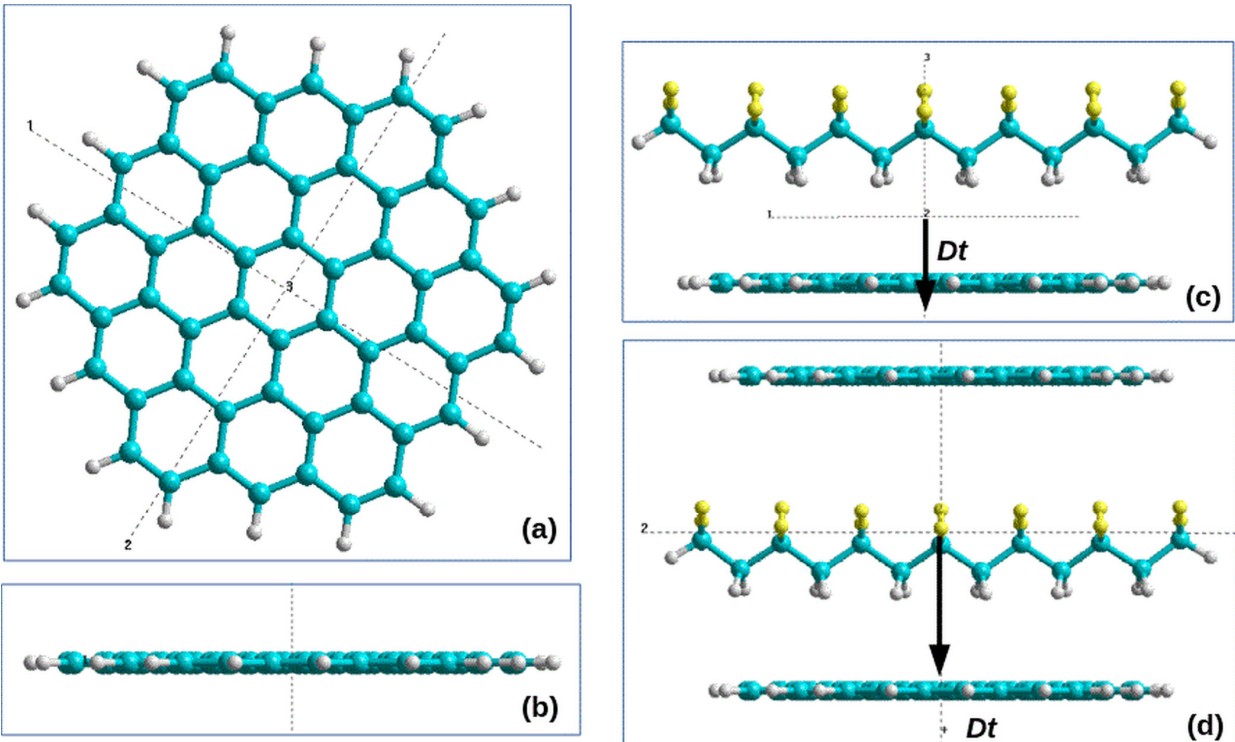

**Figure 3.** Models of initial states for graphene layers and PVDF6 chain: (**a**) one PVDF chain and one graphene layer model Gr54H with 54 carbon atoms C (cyan), surrounded by the hydrogen atoms H (gray) in the Z-plane; (**b**) the same in the Y-projection plane; (**c**) PVDF6 + Gr54H_H-C model in the initial state of the PVDF6 chain and Gr54H with "H-C" positions; (**d**) one PVDF chain and two graphene layers—the sandwich structure Gr54H + PVDF6 + Gr54H model in the initial state. Initial dipole moment *Dt* orientation is shown by a vector—the black arrow. (Reprinted with permission from [42]; Taylor & Francis 2022).

Examples of the final states of these variously considered composite models of a PVDF chain with graphene layers after such a rotation (flip/switch) are shown in Figure 4a,b (final state). Here, we can clearly see their inverted (switched) state relative to the initial states shown in Figure 3c,d (initial state). In this case, the switch occurs only with PVDF, while the graphene layers remain in the same state and form (they are only slightly distorted and shifted). The estimations of the distances between the components of this heterostructure and their change upon the switching of polarization were considered in [42].

Corresponding changes of all mean energies with time during the MD run were shown earlier in Figure 2c for pure PVDF and now in Figure 4 for both heterostructures of PVDF with graphene layers. The final rotation time of the PVDF chain (switching time $\tau_S$) was estimated similarly to that written above for these MD energy trajectories under the criteria $\delta \sim E_K / E_{Kmax} \leq 0.01$–0.001, where $E_K$ is the kinetic energy at the final point and $E_{Kmax}$ is the kinetic energy at the chain rotation maximum. As a result, the data computed allow one to obtain a linear critical behaviour of the $\tau^{-2}$ at an electric field $E \sim E_C$ at the lower limit values of this electric field for the case when $E \rightarrow E_C$ (for $E > E_C$); this also fully corresponds to Equation (1) from LGD theory (as described above in Section 3.1).

The results of changing the switching time $\tau_S$ and $\tau^{-2}$, obtained from the calculations of MD runs for various values of the electric field *E*, for both heterostructure type, are shown below in Figure 5. The scatter of the obtained data and estimates of the error in the switching time (rotation of the dipole vector *Dt*) in the electric field *E*, as determined by the relation $\delta \sim E_K / E_{Kma}x$, were analyzed previously in [42].

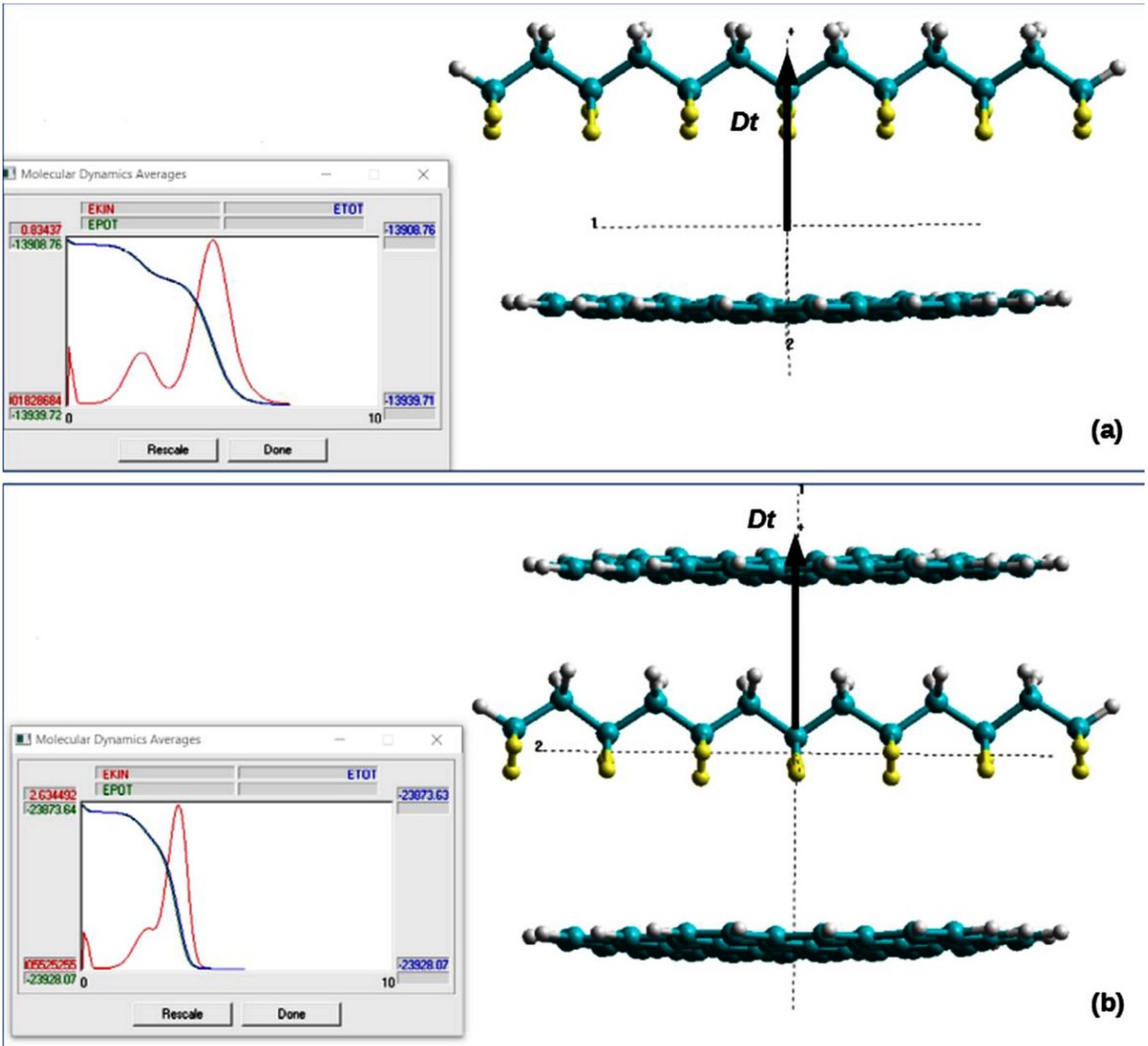

**Figure 4.** Final states of the heterostructure PVDF6 + Gr54H models after PVDF chain and total dipole *Dt* rotation (switching) in the applied electric field *E*: (**a**) one-side PVDF6 + Gr54H_H-C model with switched opposite orientation of the total dipole moment *Dt* in comparison with the initial state in Figure 3c; (**b**) sandwich Gr54H + PVDF6 + Gr54H model with the same opposite switched *Dt* in comparison with initial state in Figure 3d. The right-hand side shows the MD energy trajectories over time during the MD run. (Reprinted with permission from [42]; Taylor & Francis 2022).

In this work, new calculations were carried out and the basic average data were obtained, taking into account the errors. Figure 5 presents the final improved new data (compared to [42]), taking into account the new error analysis of MD runs and switching time calculations.

The data obtained allow us to draw the following conclusions.

First, the behavior of the PVDF chain presented here (the black line in Figure 5) compares well with previous calculations of the PVDF chain (Figure 2c) considered in [18,20,21], which showed linear behavior according to Equation (1) with a coercive field $Ec_1 \sim 0.001$ a.u.$\sim 0.5$ GV/m.

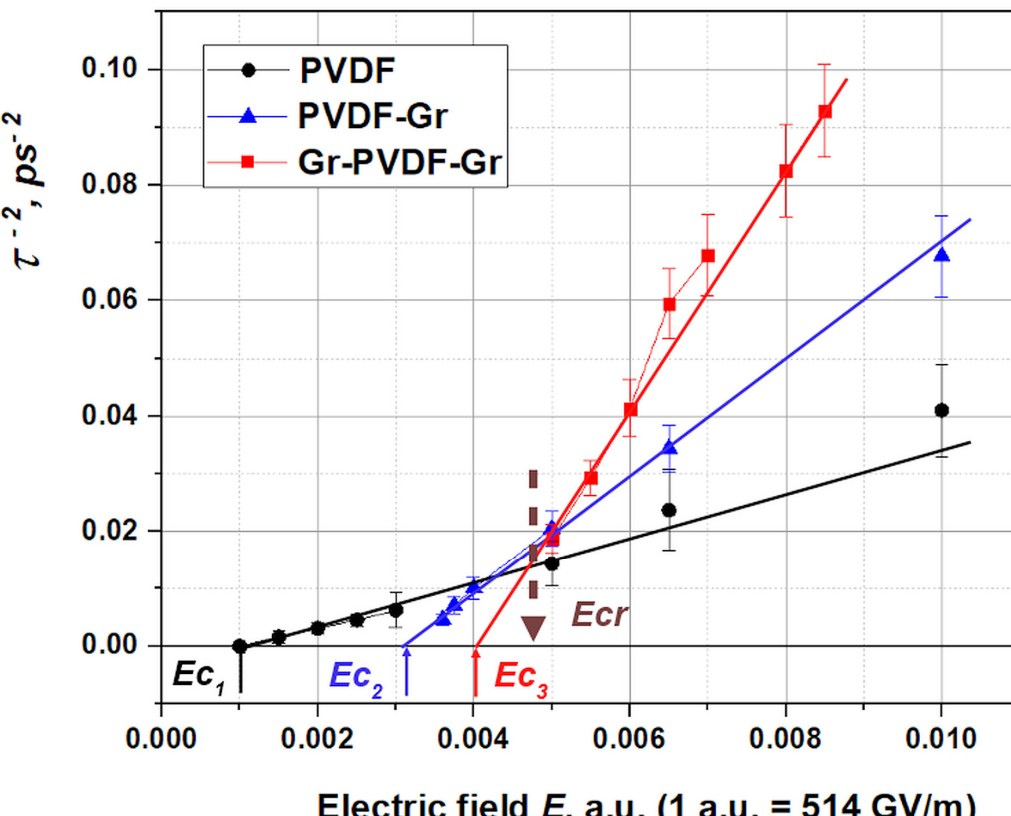

**Figure 5.** Values of the inverse square $\tau_S^{-2}$ of the switching time $\tau_S$ (1) as a linear function of the electric field $E$ in the vicinity of coercive field $E_C$ for different models. The main lines are as follows: black line—the PVDF6 (PVDF) chain model of PVDF without graphene component; blue line—the model PVDF6 + Gr54H_H-C (PVDF-Gr) with one-side graphene layer; and red line—the sandwich model Gr54H + PVDF6 + Gr54H (Gr-PVDF-Gr) with 2 graphene layers. The intersection area for all lines is at $E{\sim}Ecr$.

Second, a study of the behavior of the switching time on a model of a one-side heterostructure PVDF6 + G54H_H-C (initial state in Figure 3c, behavior of the switching time is shown by the blue line in Figure 5) indicates that the time $\tau_S$ has became shorter. This means that the motion of the dipole $Dt$ occurs faster under the action of graphene at the same electric field $E$, in the interval of increasing fields $E > Ec_2$. Meanwhile, the coercive field increases up to the value $Ec_2{\sim}0.003$ a.u.${\sim}1.5$ GV/m.

This means that in the vicinity of $Ec_2$, the switching time $\tau_S$ begins to increase more significantly than in the first case of $Ec_1$. Thus, the slowdown in the $Dt$ rotation occurs in the immediate vicinity of $Ec_2$, which is more noticeable than in Case 1. This shift in the behavior of $\tau_S$ occurs when the field $E$ in the surrounding region is less then some critical or intersection value of $E = Ecr{\sim}0.0045$ a.u.${\sim}2.3$ GV/m. This is shown as the intersection area at $E = Ecr$ in Figure 5, marked with a dashed brown arrow.

Third, the behavior of the Gr54H + PVDF6 + Gr54H sandwich heterostructure demonstrates even shorter switching times $\tau_S$ (see Figure 3d and the red line in Figure 5) and an increasing of the coercive field up to $Ec_3{\sim}2$ GV/m. In this case, the changes in the switching time $\tau_S$ occur similarly around of the point of $E = Ecr$, but it is sharply pronounced.

At the same time, the graphene layer does not significantly move or turn over but basically retains its position. It may be only a little relaxed, moved and shifted (while maintaining the general center of mass or center of inertia of the entire simulated heterostructure system). Only the PVDF chain was rotated in the applied electric field, because it had a dipole moment.

Thus, the MD run calculations performed using the semi-empirical PM3 method at each MD step clearly show that the graphene layers act on the switching times $\tau_S$ in the ferroelectric "PVDF-graphene" composite heterostructure and shift its values, depending of the applied field E and the number of graphene layers, but this is differently revealed around the vicinity of the critical *Ecr*.

In this case, there is also a change in the values of the coercive fields $Ec_1$, $Ec_2$, and $Ec_3$ of the entire ferroelectric heterostructure (with a different number of interstitial layers of graphene) as a thin nano-sized layered coating.

### 4.2.3. The Possible Reasons for the Graphene Layer Influence on Switching Times

Comparison and correlation of the obtained results with known experimental data can only be carried out based on the relative changes in these parameters (switching times) in these switching processes, since the immediate time scales of the simulated structures and experimental samples are very different. However, we can use some experimental results from [51]. Such an analysis was recently carried out in [42].

Considering the kinetics of the polarization switching using the LGD theory showed that such a change in the switching time could be issued to decrease the damping coefficient $\zeta$ value, included in relation (2), that influenced the switching time, in accordance with the Landau–Khalatnikov kinetic equations in [50–52].

Thus, it was found that the introduction of first one layer of graphene and then two layers of graphene leads to a decrease in the damping coefficient $\zeta$ compared to the initial case of a pure single layer of PVDF when it is rotated (switched) in an electric field E—these values of $\zeta$ decrease successively [42], as

$$\zeta = \zeta_1 = (3.61)10^{10} msVC^{-1},$$

$$\zeta_2 = \zeta_1/3 = (1.2)10^{10} msVC^{-1},$$

$$\zeta_3 = \zeta_1/4.47 = (0.81)10^{10} msVC^{-1}$$

Thus, the resulting changes in switching times may have a physical cause in the resulting difference in the damping (attenuation) coefficients (according to the LGV theory and the Ladau–Khalatnikov equation) when graphene layers are introduced into the heterostructure of the ferroelectric coating, due to their induced electrostatic interaction with the dipole PVDF/PVDF-TrFE chain of the ferroelectric polymer.

### 4.3. Polarization Switching in PVDF-TrFE

#### 4.3.1. Main Details

To carry out similar MDS runs with polymer ferroelectrics based on PVDF-TrFE copolymers, we also chose here a simpler model of six main units of the polymer chain. The fact is that longer chains, when turning in strong electric fields, often "get entangled" and cannot reach an equilibrium final state for a long time. This distorts the results of determining the switching time. In addition, the correctly chosen initial symmetry of the model structure is important here, since the rotation of the chain occurs under conditions of "free suspension" in space while maintaining the general center of mass or the value of inertia of the entire system. If the initial configuration is poorly chosen, the mutually consistent motion of all atoms of the system can lead to noticeable distortions in the final configuration of the structure.

In this case, we chose a symmetrical initial arrangement at the positions of fluorine atoms, which replace hydrogen atoms in the copolymer structure (Figure 6a). This made it possible to conduct adequate numerical experiments on MD runs at different values of the applied electric field and obtain a well-symmetrical rotation of the copolymer chain to the final state (Figure 6b). It should be noted that this model is quite close to experimental structures, with ratios of hydrogen and fluorine atoms at the level of (70:30) and (60:40), which are usually used in experiments [19,22–25].

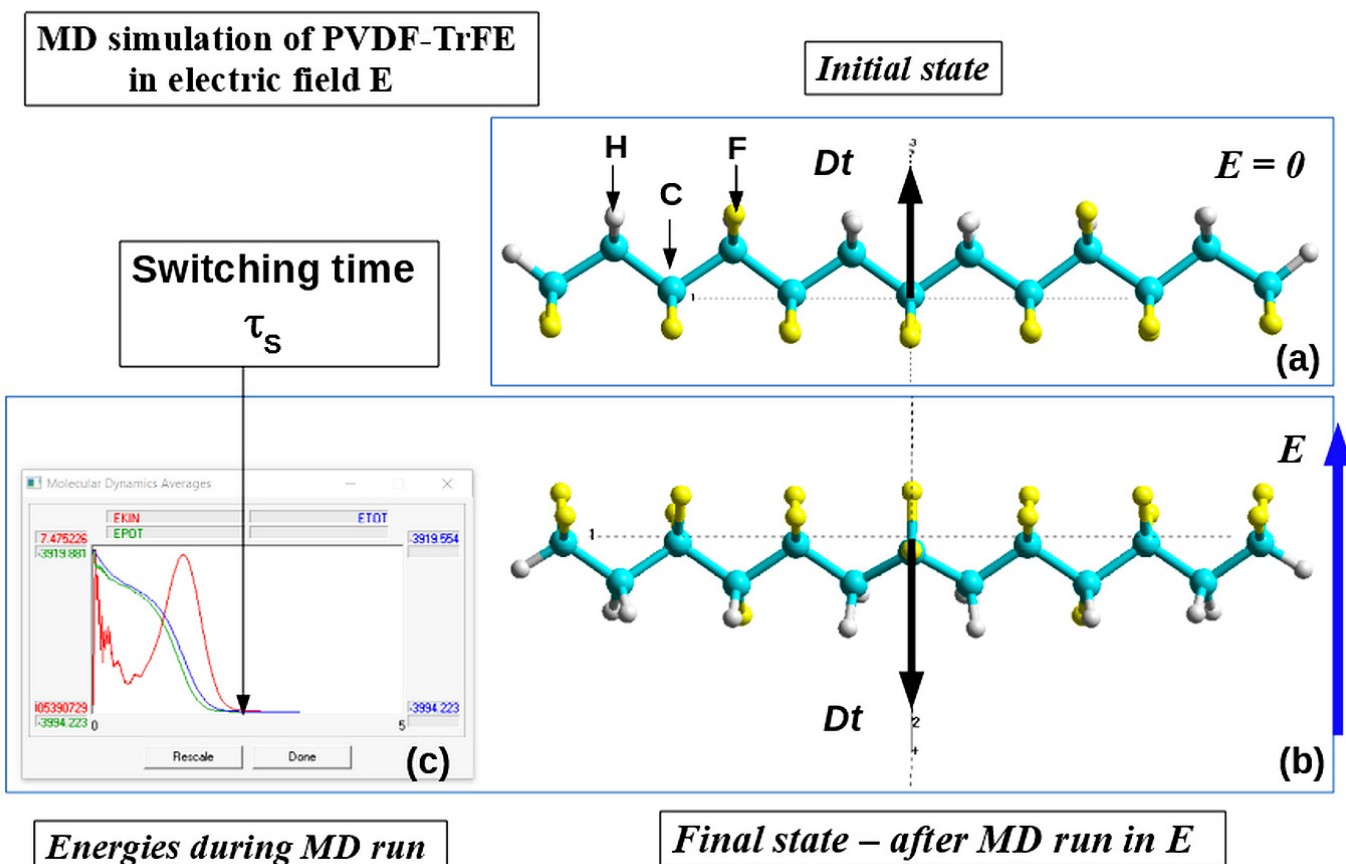

**Figure 6.** Schematic of the MD run process in applied electric field *E* for one copolymer PVDF-TrFE chain (PVDF6-TrFE model) using PM3 RHF method calculations by HyperChem software at each MD run step: (**a**) initial state and (**b**) final state after MD run with rotation (switching) on the opposite direction total dipole moment $D_t$ orientation in electric field *E*. Inset (**c**) show the energy changes during MDS run with respect to time (in ps). The arrows show the dipole moment vector orientation.

4.3.2. Results

MDS runs at different applied values of the external electric field *E*, performed similarly and according to the method described above (Section 3.1), made it possible to obtain the polarization switching times $\tau_S$ depending on the field *E* for this copolymer model case.

On this basis, similar graphs of the dependence of the square of the reverse switching time $\tau_S^{-2}$ were constructed on the magnitude of the electric field *E*, which also turned out to be linear (especially near the values of the corresponding coercive field *Ec*), in full accordance with Equation (1). Figure 7 presents these results, from which the linear law of behavior $\tau_S^{-2}$ on *E* is clearly visible, and the position and value of the coercive field $Ec_1{}^* = \sim0.002$ a.u.~1.028 GV/m for PVDF-TrFE is easily determined.

The results obtained clearly show that the switching time $\tau_S$ in PVDF-TrFE becomes shorter compared to pure PVDF, while the coercive field $Ec_1{}^*$ becomes larger in comparison with $Ec_1$ for PVDF, and its values correspond to both experimental data [19] and values previously calculated [39,40] by various methods.

*4.4. Polarization Switching in a Heterostructure Consisting of PVDF-TrFE and Graphene Layers*

4.4.1. Main Details

MD runs for cases of PVDF-TrFE heterostructures with both types of graphene layer models (one-sided model and two-sided sandwich model) are considered here in the same way, as described above for cases with PVDF.

The initial states of both heterostructure models are shown in Figures 8 and 9.

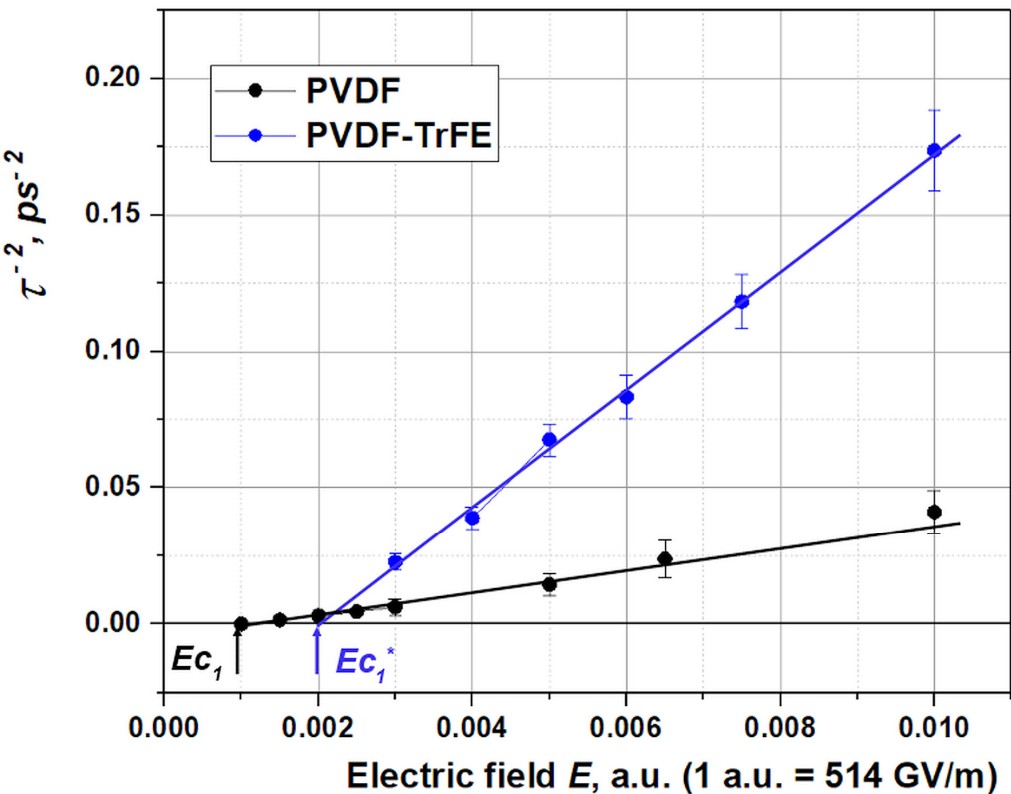

**Figure 7.** Values of the inverse square $\tau_S^{-2}$ of the switching time $\tau_S$ (1) as a linear function of the electric field $E$ in the vicinity of coercive field $E_C$ for PVDF-TrFE in comparison with PVDF models. The main lines are as follows: black line—the PVDF6 chain model of PVDF; blue line—the model PVDF6-TrFE model.

4.4.2. Results

The main results obtained for these models are shown in Figures 10 and 11.

Analysis of the obtained results of calculations of the polarization switching time on the applied electric field for models of heterostructures based on polymer ferroelectrics and graphene layers, as presented in Figures 10 and 11, allows us to draw the following conclusions:

(1)  In all cases, for all considered models of both the initial polymer ferroelectric PVDF and PVDF-TrFE copolymer, and for their heterostructures with graphene layers, the linear behavior of the square of the reciprocal switching time $\tau^{-2}$ from the magnitude of the applied electric field $E$ is demonstrated, in good agreement with Equation (1) at field values greater than the coercive field $E > Ec$ and in the immediate vicinity of $Ec$, which is quite consistent with expression (1) obtained from the Landau–Khalatnikov equation in works [50–52], and it fully corresponds to the LGD theory for such thin homogeneous layers of ferroelectrics [20,21].

(2)  The influence of graphene layers, in the case of a one-sided model, leads to an increase in $\tau^{-2}$ values depending on $E$ and, accordingly, to a decrease in the switching time $\tau_S$, both in the cases of pure PVDF and its PVDF-TrFE copolymer; in this case, the values of the coercive field are shifted: in the case of changes from PVDF to PVDF-Gr, the value of $Ec_1 \sim 0.001$ a.u.$\sim 0.5$ GV/m increases to $Ec_1^* \sim 0.002$ a.u.$\sim 1$ GV/m, whereas in the case of changing PVDF-TrFE to PVDF-TrFE-Gr, the coercive field decreases from $Ec_2 \sim 0.003$ a.u.$\sim 1.54$ GV/m to $Ec_2^* \sim 0.0018$ a.u.$\sim 0.92$ GV/m.

(3)  The influence of graphene layers in the case of a two-sided (sandwich) model is more complex: if initially, when pure PVDF is included between two layers of graphene, there is a further increase in $\tau^{-2}$ depending on $E$ (and, accordingly, a decrease in switching time $\tau_S$) with increasing coercive field to values of $Ec_3 \sim 0.004$

a.u.~2.06 GV/m, then in the case of the PVPD-TrFE copolymer, these changes occur differently: after increasing the $\tau^{-2}$ values depending on $E$ for the one-sided PVDF-TrFE-Gr model, for the sandwich model Gr-PVDF-TrFE-Gr, there is a noticeable decrease in this dependence so that the $\tau^{-2}$ values on $E$ become even smaller than in the case of one PVDF-TrFE (that is, the switching times $\tau_S$ themselves increase) and the coercive field increases to values of $Ec_3$* = 0.0032 = 1.65 GV/m.

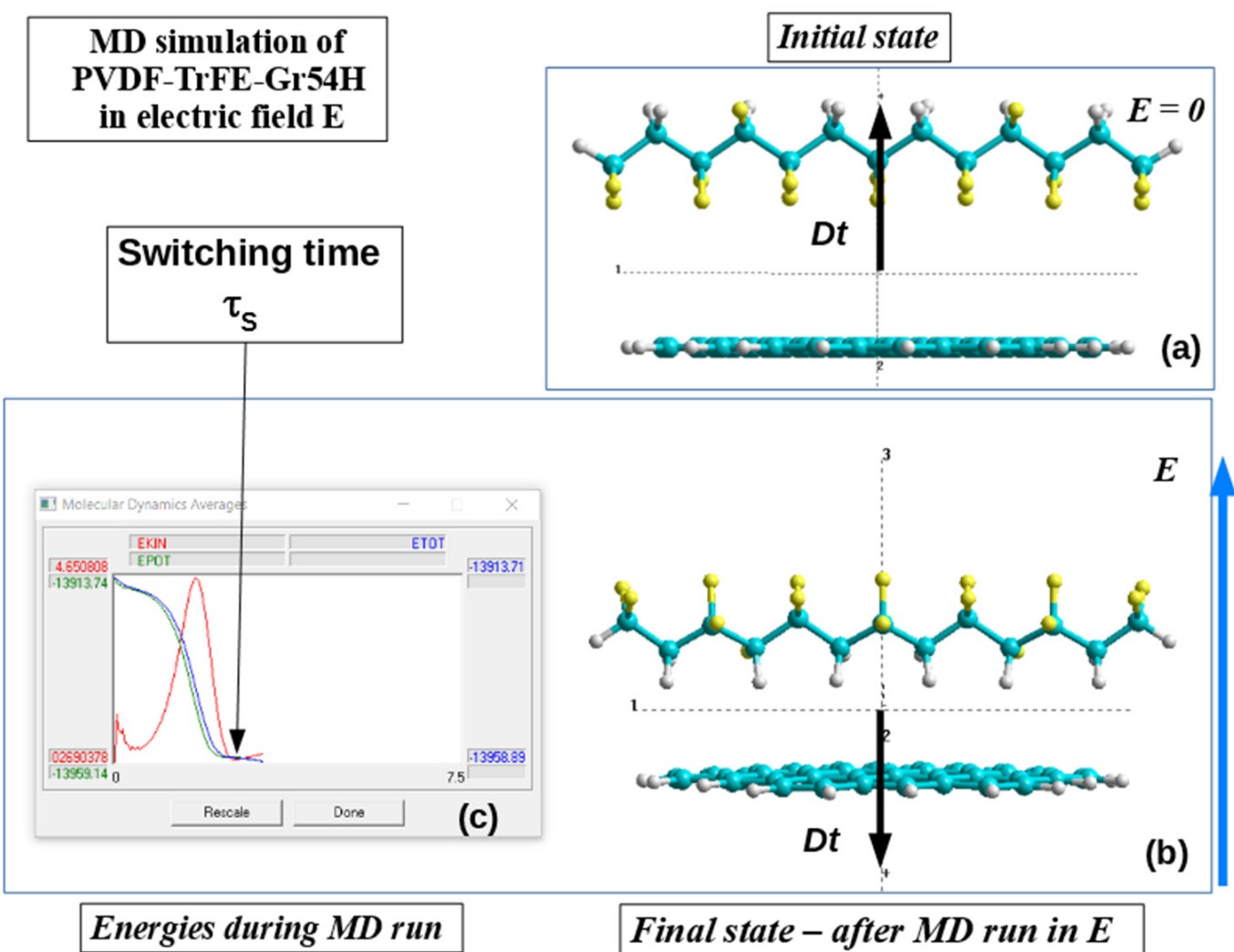

**Figure 8.** Models of graphene layers and PVDF-TrFE chain: (**a**) initial states for PVDF6-TrFE model chain and one graphene layer Gr54" model with 54 carbon atoms C, surrounded by the hydrogen atoms H; (**b**) final state of PVDF6 + G54H_H-C model after rotation in electric field E during MD run; and (**c**) energy trajectories. Dipole moment *Dt* orientation is shown by the black arrow.

The obtained values of switching times and coercive fields for the copolymer correspond well to the known experimental values, but for models with graphene they were obtained here for the first time and these calculated and predicted values can serve as a good guide for new experimental studies in this area when creating such promising heterostructures based on thin polymer layers of ferroelectrics and graphene.

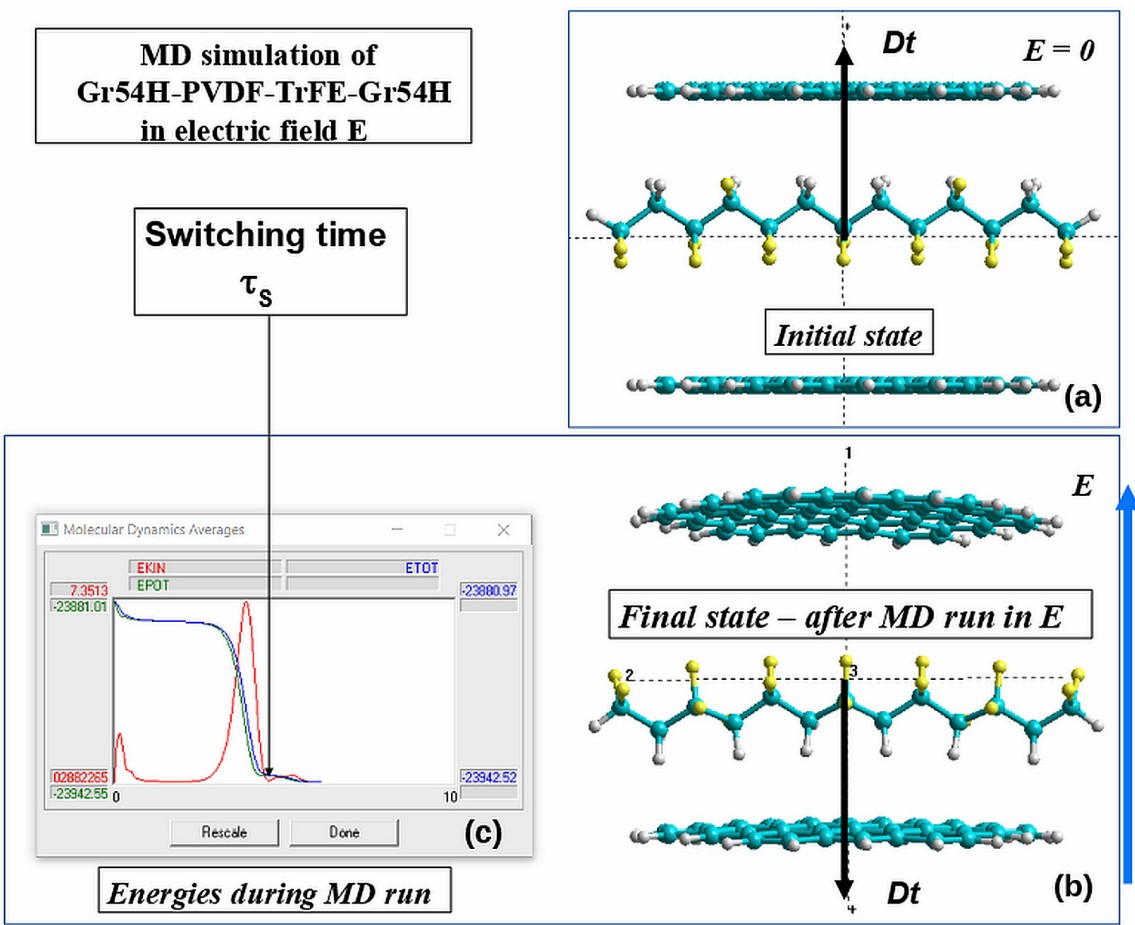

**Figure 9.** Models of 2 graphene layers and PVDF-TrFE chain: (**a**) initial states for PVDF6-TrFE model chain and 2 graphene layer Gr54H model with 54 carbon atoms C, surrounded by the hydrogen atoms H; (**b**) final state of Gr54H-PVDF6 + G54H model after rotation in electric field E during MD run; and (**c**) energy trajectories. Dipole moment *Dt* orientation is shown by the black arrow.

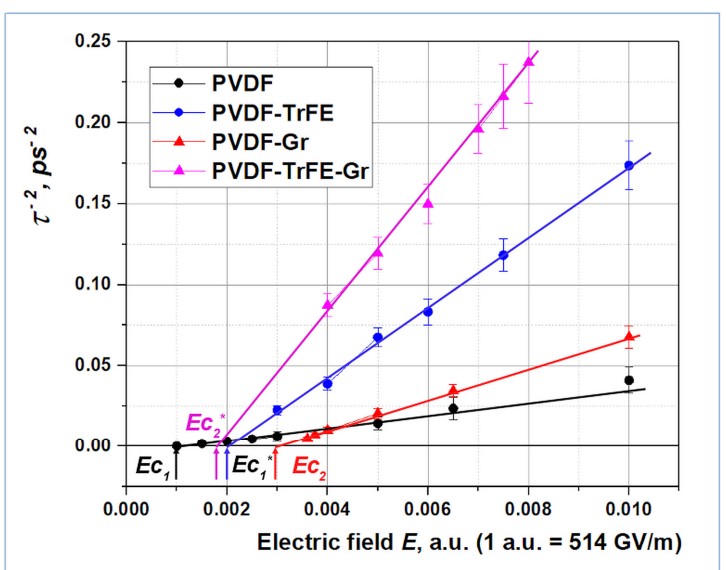

**Figure 10.** Values of the inverse square $\tau_S^{-2}$ as a linear function of the electric field *E* for PVDF and PVDF-TrFE in comparison with PVDF-Gr and PVDF-TrFE-Gr models. Lines are as follows: black—PVDF; blue—PVDF6-TrFE; magenta—PVDF6-Gr54H; red line—PVDF6-TrFE-Gr54H.

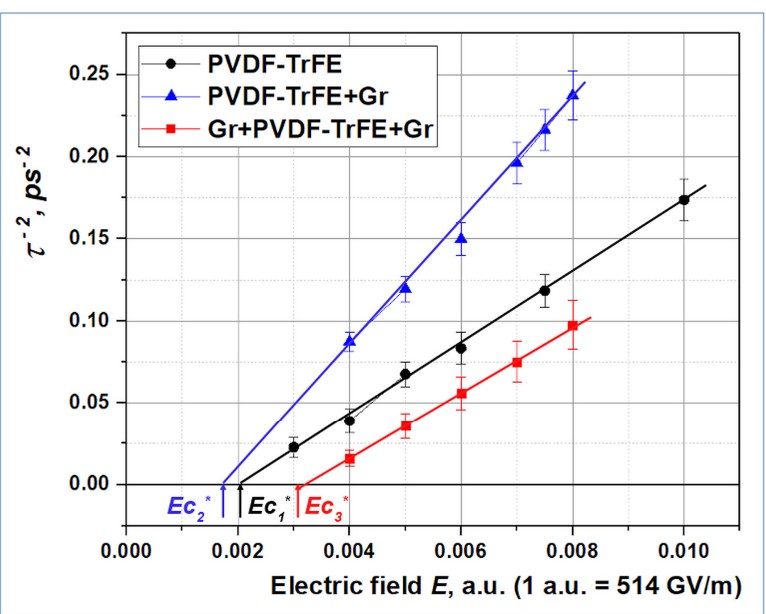

**Figure 11.** Values of the inverse square $\tau_S^{-2}$ as a linear function of the electric field $E$ for PVDF-TrFE and PVDF-TrFE-Gr in comparison with sandwich Gr-PVDF-TrFE-Gr models. Lines are as follows: black—PVDF6-TrFE; blue—PVDF6-TrFE-Gr54H; red line—Gr54H-PVDF6-TrFE-Gr54H.

## 5. Conclusions

We presented a modeling and computational study of a composite hybrid nanostructure containing a ferroelectric PVDF/PVDF-TrFE polymer, and graphene-like layers were carried out in this work using molecular dynamics (MD) simulations with a semi-empirical quantum PM3 method in the Hartree–Fock approximation (RHF) from the HyperChem software. The calculations and MD run performed in this article unambiguously show that the coercive field increases and the polarization switching time mainly decreases under the influence of graphene layers in thin layers of polymer ferroelectrics. However, in the closest vicinity of the coercive fields $Ec_i < Ecr$, below the intersection point $Ecr$ of the dependences $\tau_S(E)$, the behaviour of the switching times is reversed, and they increase here more sharply under the influence of the embedded graphene layers.

The obtained results of modeling PVDF-TrFE copolymers and MDS runs on these models showed that they have switching times shorter than pure PVDF. This is important for practical applications. In addition, it has been shown that the introduction of a single graphene layer further reduces the switching time in such a heterostructure.

At the same time, in the case of a double-sided (sandwich) model with two layers of graphene around a PVDF-TrFE layer, switching times increase. This is also an undoubtedly important result that must be taken into account when creating practical heterostructures based on graphene and polymer ferroelectrics.

As a result of the MDS calculations performed, it is unambiguously shown that for all considered thin ferroelectric heterostructures based on polymer ferroelectrics and graphene, the LGD theory is completely valid; it well describes the linear law of the behavior of the square of the inverse polarization switching time in the immediate vicinity of the coercive field, i.e., homogeneous switching in thin ferroelectric structures. Analysis of the kinetics of the polarization switching within the framework of the LGD theory showed that this is possible due to a decrease of the damping coefficient, describing switching time in the Landau–Khalatnikov kinetic equations and LGD theory.

These results obtained using computational modeling and calculations, of course, require deep experimental verification and further investigations. This can be performed, for example, by production of the coated samples by PVDF layers deposited directly (using the LB method) on the graphene-like layers or substrate hetero-structures (of different thicknesses and/or different numbers of layers—similar, for example, to those proposed and

discussed in [20,21,41,42]). Following measurements, the switching times and hysteresis loops (such as in the coating works performed in [23–25,41,54]) of these samples could be performed for experimental verification. The results of the performed investigation are important and actual, since they demonstrate the path of the possible modifications of new nanomaterials and coatings with the predetermined and necessary properties. Based on these computed results, new hybrid heterostructures and coatings of various nanomaterials with adjustable polarization switching times and coercive fields can be fabricated using polymer ferroelectrics and graphene/graphene oxide layers and similar two-dimensional hybrid coating heterostructure. This opens up new opportunities for creating a new generation of nanodevices and sensors.

**Author Contributions:** E.P. and V.B. wrote the manuscript. V.F. supervised and supported this study and critically revised the manuscript. X.M., H.S., T.L. and J.W. contributed to experiments and data analysis. All authors have read and agreed to the published version of the manuscript.

**Funding:** This research received no external funding.

**Institutional Review Board Statement:** Not applicable.

**Informed Consent Statement:** Not applicable.

**Data Availability Statement:** The data presented in this study are available on request from the corresponding author.

**Acknowledgments:** Xiang-Jian Meng expresses his gratitude to the National Natural Science Foundation of China (NNSFC) for grant # 61574151 and Hong Shen for grant # 62011530043. The authors are grateful for the opportunity to perform calculations using the computing and information resources of the IMPB RAS.

**Conflicts of Interest:** We declare no potential conflicts of interest in this article.

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
