# Peer review of "Ferroelectric Thin Films and Composites Based on Polyvinylidene Fluoride and Graphene Layers: Molecular Dynamics Study"

_coatings, doi:10.3390/coatings14030356_

Round 1
Reviewer 1 Report
Comments and Suggestions for Authors
The article is about the molecular dynamic simulation of thin film composites. However, there are several things that could be done to improve the paper.
1. The authors must take into account the interests of the readers of Coatings MDPI journal. The authors must add the implications of the study for coating purposes. This should be written in abstract and conclusion at least. Currently, it contains too much computational chemistry.
2. Please revise the section and subsection numbering. It should be 2. Basic models and methods
3. Graphene tends to agglomerate or restack during drop casting. At some point, it will behave like graphite. Could you compare the result with graphite as well? It would be interesting to see at which point it will behave like graphite instead of graphene.
4. It would be better to include the schematic of each sample to better visualize a one sided and two sided sandwich model.
Author Response
Please see the attachment
Dr. Vladimir Bystrov

Reviewer 2 Report
Comments and Suggestions for Authors
The work should be reformatted and the results should be presented more clearly and concisely.
Manuscript needs to be corrected in English and technically arranged (spacing between words, labels on graphics, chapter headings).
Some of the concrete proposals:
Title should be changed...Molecular Dynamics Simulation Studies of Properties...
The introduction should be written more systematically... although the authors provide the necessary information in the introduction (why MDS is important, what is the application of the examined polymer and composite, what is the topic of the paper), they should be systematized into logically consequent units.
Text in the whole paragraph 2.2. are too extensive, shorten them
Too many pictures that are reprinted in the Results paragraph, this is not a review paper and they additionally burden the manuscript.
Mark the axes on the graphs in Figure 4
The heading of paragraph 3 and sub-heading 3.3.2 is the same Results
In the introduction (page 2, line 75) it is written ". It is important that these theoretical 75 studies were carried out in close combination with their experimental studies." In the paper and conclusion, it is not clear to which experimental confirmations it refers.
​
Author Response
Please see attachment
Dr. Vladimir Bystrov

Reviewer 3 Report
Comments and Suggestions for Authors
This work presents a molecular dynamics simulation study on ferroelectric polymer thin films and their composites with graphene layers. The research focuses on understanding the polarization-switching dynamics of polyvinylidene fluoride (PVDF) and its composites under various conditions. The MD simulations aim to explore the influence of graphene layers on the switching times and the coercive field of the materials. The study integrates quantum-chemical semi-empirical methods and the Hartree-Fock approximation to predict the behavior of these nanostructured materials.
Some points should be considered properly/improved/added:
1. Improve all images by increasing resolution and labeling font.
2. Along with the methodology, there is an excessive number of references, and it is difficult to identify the novelty proposed in the present study. Modeling strategy, systems under consideration...? Seems like everything that's been done before.
3. The LGD theory is foundational for their work, but the authors should critically assess its applicability to nanoscale systems, as it traditionally describes bulk properties. Deviations at the nanoscale can occur, and the models may need adjustments. The link to Refs. do not reflect the direct applicability to the current investigation. The LGD theory is a phenomenological theory traditionally used for bulk materials. Its application to two-dimensional ferroelectric systems needs further context. The coefficients α, β, and γ in the LGD theory have temperature and pressure dependence.
4. Both the LGD free energy expansion and the Landau-Khalatnikov equation assume a homogeneous material. However, thin films can have gradients in composition, defects, and other inhomogeneities that impact ferroelectric properties.
5. To further clarify, the gradient term is mentioned as potentially being included in the general case. If spatial variations in polarization are significant, this term cannot be neglected.
6. An exhaustive Discussion section should be included to contextualize the observed findings. In fact, Section 2 is simply called Results and the description is limited to what was observed.
7. The results for materials with graphene layers should be interpreted and discussed in detail because a high-quality material is assumed. The physical mechanisms by which graphene influences the ferroelectric properties should be elucidated, as this could be due to a variety of factors such as mechanical reinforcement, electrical conductivity changes, or interface effects. The results for the copolymer are said to correspond well with known experimental values, which is good. However, for the models with graphene, since the values are obtained for the first time, it will be crucial to validate these findings with independent experimental studies.
8. The comparison between pure PVDF, its copolymer, and the composites with graphene should be contextualized in terms of their applications. The implications of the observed changes in switching times and coercive fields on the performance of devices using these materials need to be discussed properly.
Comments on the Quality of English Language
Some typos here and there.
As an example, "molecular MDS runs" which is read as "molecular molecular dynamic simulations", "Let's" contraction is not commonly used, and so on...
Author Response

(The authors gave the same response as above.)

Round 2
Reviewer 1 Report
Comments and Suggestions for Authors
The authors have addressed all the comments
Reviewer 2 Report
Comments and Suggestions for Authors
The manuscript may be accepted for publication.
Reviewer 3 Report
Comments and Suggestions for Authors
Thank you for considering comments and suggestions.
I recommend the publication of the revised version.